# Sustainable upcycling of mixed spent cathodes to a high-voltage polyanionic cathode material

Guanjun Ji [1,2,5], Di Tang[1,5], Junxiong Wang [1,2] ✉, Zheng Liang [2], Haocheng Ji[1], Jun Ma[1], Zhaofeng Zhuang [1], Song Liu[1], Guangmin Zhou [1] ✉ & Hui-Ming Cheng [3,4] ✉

Sustainable battery recycling is essential for achieving resource conservation and alleviating environmental issues. Many open/closed-loop strategies for critical metal recycling or direct recovery aim at a single component, and the reuse of mixed cathode materials is a significant challenge. To address this barrier, here we propose an upcycling strategy for spent $LiFePO_4$ and Mn-rich cathodes by structural design and transition metal replacement, for which uses a green deep eutectic solvent to regenerate a high-voltage polyanionic cathode material. This process ensures the complete recycling of all the elements in mixed cathodes and the deep eutectic solvent can be reused. The regenerated $LiFe_{0.5}Mn_{0.5}PO_4$ has an increased mean voltage (3.68 V versus Li/Li$^+$) and energy density (559 Wh kg$^{-1}$) compared with a commercial $LiFePO_4$ (3.38 V and 524 Wh kg$^{-1}$). The proposed upcycling strategy can expand at a gram-grade scale and was also applicable for $LiFe_{0.5}Mn_{0.5}PO_4$ recovery, thus achieving a closed-loop recycling between the mixed spent cathodes and the next generation cathode materials. Techno-economic analysis shows that this strategy has potentially high environmental and economic benefits, while providing a sustainable approach for the value-added utilization of waste battery materials.

The sustainable recycling of lithium-ion batteries (LIBs) has gradually become a focus of attention in recent years[1–3]. Among all the components involved in a battery, cathode materials account for the largest mass and dominate the battery cost[4,5]. In this context, the recycling of cathode materials is very important due to the environmental and economic benefits. Commercial cathode materials include olivine $LiFePO_4$ (LFP), layered $LiCoO_2$ (LCO) or $LiNi_xMn_yCo_{1-x-y}O_2$ (NMC), and spinel $LiMn_2O_4$ (LMO)[6]. Current recycling methods primarily target one specific cathode material and

it is challenging to simultaneously process multiple compositions due to their different crystal structures. Traditional pyrometallurgical and hydrometallurgical recycling methods destroy the battery cells or materials by high-temperature melting or chemical processes and require complex steps to separate the different elements, resulting in secondary pollution[5,7,8]. More importantly, the amount of valuable elements in spent LFP (S-LFP) or LMO (S-LMO) is small, and the relatively low economic value of Fe/Mn products produced by these methods means that the current recycling

[1]Tsinghua Shenzhen International Graduate School, Tsinghua University, Shenzhen 518055, China. [2]Frontiers Science Center for Transformative Molecules, School of Chemistry and Chemical Engineering, Shanghai Jiao Tong University, Shanghai 200240, China. [3]Institute of Technology for Carbon Neutrality / Faculty of Materials Science and Engineering, Shenzhen Institute of Advanced Technology, Chinese Academy of Science, Shenzhen 518055, China. [4]Shenyang National Laboratory for Materials Science, Institute of Metal Research, Chinese Academy of Sciences, Shenyang 110016, China. [5]These authors contributed equally: Guanjun Ji, Di Tang. ✉e-mail: wjx1992@sjtu.edu.cn; guangminzhou@sz.tsinghua.edu.cn; hm.cheng@siat.ac.cn

approaches are unsuitable. Hence, better recycling techniques are needed to achieve the maximum sustainability of the mixed cathode materials.

LFP batteries have been widely used in energy storage stations and electric vehicles due to their good thermal stability and low manufacturing cost[4,9,10]. However, traditional LFP cells have a specific energy of ~180 Wh kg$^{-1}$, much lower than the >250 Wh kg$^{-1}$ of NMC and nickel−cobalt−aluminum (NCA) cells[11,12]. This difference in energy density has been much narrowed at the pack level by simplifying the battery module and improving volume utilization efficiency, such as by the emerging cell-to-pack and cell-to-body/chassis technologies[13,14]. These techniques have almost reached their limit, and a further improvement of the performance requires an increase in the specific energy of LFP itself, for which increasing the mean voltage is the most effective way[15,16]. Among olivine-type cathode materials, LFP has a limited working potential of 3.4 V versus Li/Li$^+$ compared with 4.1, 4.8, and 5.2 V for LiMnPO$_4$, LiCoPO$_4$, and LiNiPO$_4$, respectively[17–19]. However, these alternative "high-voltage" olivine-type cathodes are far from practical, either because of their poor electrical conductivity or they operate beyond the stable voltage range of the electrolyte. Substituting a transition metal (TM) at Fe$^{2+}$ sites in LFP to form a LiFe$_x$TM$_{1-x}$PO$_4$ (such as LiFe$_x$Mn$_{1-x}$PO$_4$, LFMP) solid solution is considered a promising strategy to increase the working potential[20–22]. Spent LIBs rich in these transition metals are important resources and directly upgrading S-LFP and Mn-rich cathode materials to a high-voltage LFMP cathode seems to be a feasible and economical way. This strategy not only solves the problem of recycling mixed cathode materials but also achieves upcycling of the regenerated products[23,24].

To achieve this, changing the compositions and structures of the mixed cathode materials could be a significant factor. S-LFP and S-LMO have different crystal structures, making them incompatible with the direct synthesis of polyanion-type LFMP for the following reasons. First, recycling using direct repair/regeneration approaches focuses on controlling the components and recovering the structure, but they are difficult to replace the transition metals in large quantities[25,26]. Second, the homogeneous distribution of iron and manganese in the bulk phase is a critical challenge for the performance of LFMP[27,28]. Hence, the breakdown and recombination of the mixed materials are necessary in the upcycling process. The choice of solvent needs to consider its functionality, in addition, the cost and recyclability are important factors. Deep eutectic solvents (DESs) have emerged as a class of green solvents in the last 20 years, that have the advantages of being cheap, easy to prepare, and being recyclable[29]. In recent years, DESs were used as an effective leaching regent to extract valuable elements from spent LIBs or as a lithium-ion carrier to repair the structure of cathode materials[30–32]. Thus, selecting a suitable DES with the ability to both dissolve and restructure mixed cathodes is critical.

Herein, we report an upcycling strategy for mixed spent cathode materials (S-LFP + S-LMO) which upgrades them to a high-voltage polyanionic cathode material with increased value and energy density. Specifically, a DES composed of choline chloride (ChCl) and oxalate (OA) was chosen as a regulation carrier for the composition and structure of the mixed materials. Different from traditional approaches, this strategy can achieve simultaneous dissolution and recombination of TMs, thus resulting in a solid solution with uniform distributions of Fe and Mn. Detailed characterizations revealed the structural information of regenerated LFMP (R-LFMP). R-LFMP has a mean voltage (3.68 V versus Li/Li$^+$) and a specific energy density of 559 Wh kg$^{-1}$, which is higher than that of a commercial LFP (C-LFP) cathode material (3.38 V and 524 Wh kg$^{-1}$). Meanwhile, lithium and phosphorus were recycled as a lithium salt and the DES can be reused. In addition, our proposed method is also applicable to spent LFP and LCO for the synthesis of a LiFe$_x$Co$_{1-x}$PO$_4$ (LFCP) cathode. Techno-economic analysis suggests that the upcycling strategy has environmental and economic benefits, providing a feasible and

scalable approach to upgrade mixed battery materials for the production of next-generation phosphate cathodes.

## Results

### Upcycling strategy for mixed cathode chemistries

To simulate the actual battery recycling processes, we chose S-LFP and S-LMO black mass as raw materials, as shown in Fig. 1a and Supplementary Fig. 1. LFP has an olivine crystal structure and that of LMO is spinel type, making them incompatible for the direct synthesis of LFMP with an olivine structure. Considering that substituting the Fe with Mn is the critical factor in forming a solid solution, the breakdown and recombination of the mixed materials are necessary in the upcycling process (Fig. 1b). C-LFP cathodes have the advantages of good cycling stability and sustainability but suffer from low energy density. This upcycling strategy achieves the transformation of mixed cathode materials into high-performance and value-added materials, as depicted in Fig. 1c. A recyclable DES, consisting of ChCl and OA, was used in our recycling process. Their Fourier transform infrared (FTIR) spectra show characteristic infrared peaks (Supplementary Fig. 2), and after mixing and heating, they form hydrogen bonding interactions, resulting in a reduced ability to crystallize, so that the DES formed is in a stable liquid state (Fig. 1d). This structure also improves the delocalization of the hydrogen protons in oxalic acid, increasing the acidity of the DES[29,33]. As a result, olivine S-LFP and spinel S-LMO cathode materials could be efficiently and simultaneously dissolved at a relatively low temperature of 110 °C (Fig. 1e).

The proposed dissolution mechanism is elucidated as follows (Fig. 1b): hydrogen in the OA molecules attacked the olivine structure of LFP and the spinel structure of LMO by destroying the Li-O and Me-O (Me = Fe/Mn) bonds. Then the (C$_2$O$_4$)$^{2-}$ or ChCl reduced Me$^{3+/4+}$ to Me$^{2+}$, further promoting the leaching of Me and resulting in the collapse of the crystal structure. Li and Me dissolved in the DES combined with chlorine in ChCl to form stable [LiCl$_2$]$^-$ and [MeCl$_4$]$^{2-}$ complexes[34,35]. Furthermore, the introduction of water as a diluent decreased the inherently high viscosity of the DES, thus transforming the [MeCl$_4$]$^{2-}$ complexes to MeC$_2$O$_4$·2H$_2$O precipitation[31,36]. The isolated Li$^+$ and PO$_4^{3-}$ were recycled as a lithium salt and the DES was reused for the next loops. The FTIR analysis of the original DES showed −OH and C−H stretching vibration peaks at 3291 and 3023 cm$^{-1}$ respectively, and peaks at 1710 and 864 cm$^{-1}$ were respectively ascribed to C=O and N−H bending vibrations[36] (Fig. 1f). The recycled DES had the same infrared peaks and could be reused to synthesize the precursors (Supplementary Fig. 2) because only the oxalate was consumed. The regeneration process of the DES is described in the Methods section. The inductively coupled plasma-optical emission spectrometry (ICP-OES) results show that our strategy has high leaching and separation efficiencies for all elements (Fig. 1g and Supplementary Table 1). 89.6% of the Fe and 82.2% of the Mn were precipitated in the solid solution precursor. While 96.7% of the Li and 97.0% of the P remained in the filtrate, which were then recycled as a lithium salt, Li$_3$PO$_4$, as confirmed by the X-ray diffraction (XRD) pattern (Supplementary Fig. 3).

XRD patterns of FeC$_2$O$_4$·2H$_2$O and MnC$_2$O$_4$·2H$_2$O respectively correspond to those standard peaks of iron and manganese oxalate (Fig. 1h). But the XRD pattern of the (Fe, Mn)C$_2$O$_4$·2H$_2$O precursor is not a simple superposition of two single oxalates, suggesting that the iron-manganese precursor is a mixed oxalate on the atomic scale rather than a mechanical mixture of two separate oxalate phases. This was confirmed by thermogravimetric analysis (TGA), as shown in Supplementary Fig. 4. The thermal decomposition stages of iron and manganese oxalates show different mass loss steps, because the onset decomposition temperature is related to the electronegativity of centering metal ions of the oxalates[31,37,38]. Only two mass loss steps occur for the (Fe, Mn)C$_2$O$_4$·2H$_2$O precursor, suggesting it is a solid solution. Scanning electron microscope (SEM) images show a rod-like

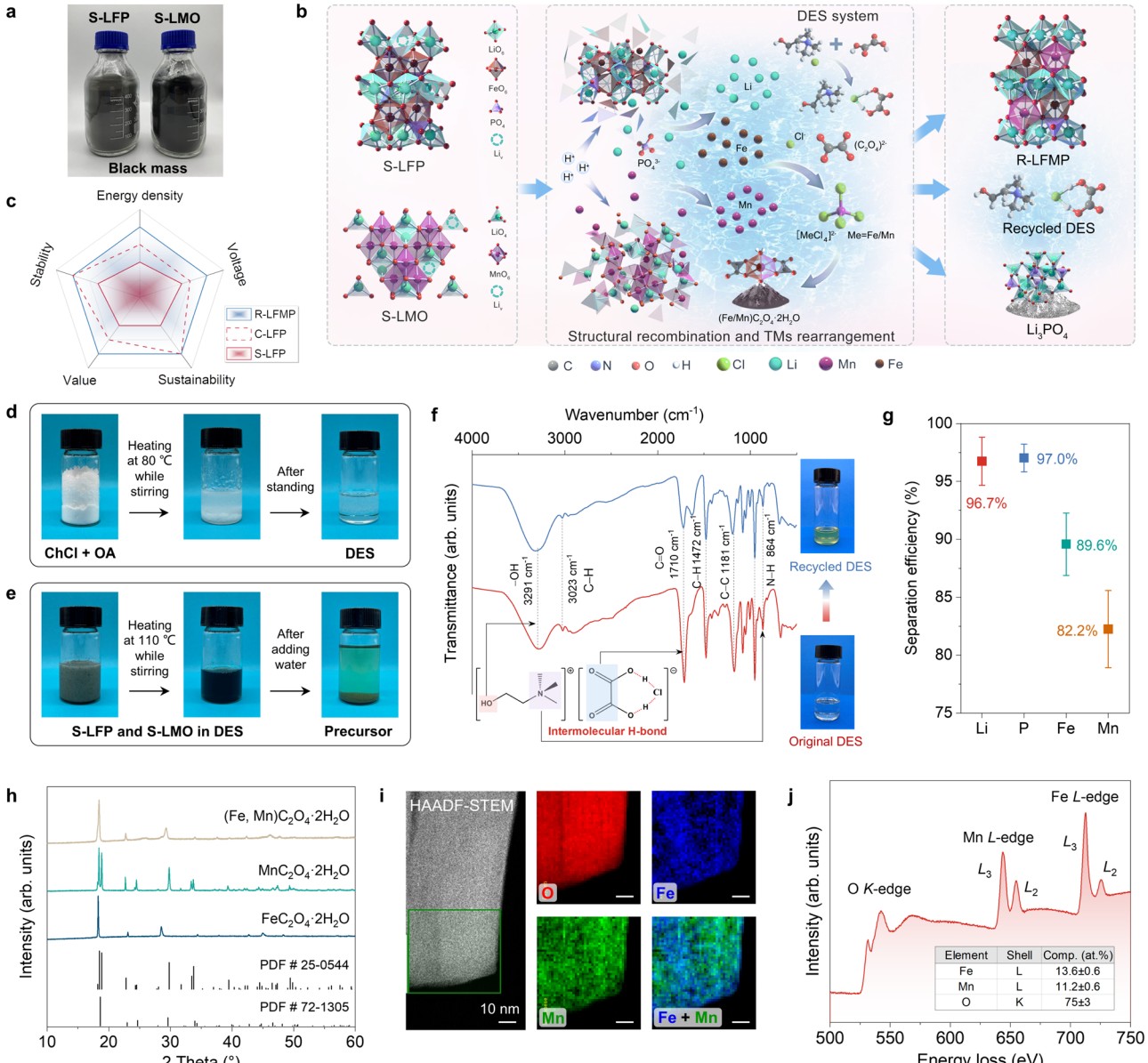

**Fig. 1 | The upcycling strategy for S-LFP and S-LMO to achieve the recovery of all elements and the synthesis of solid solution precursors. a** Photos of S-LFP and S-LMO black mass used in this work. **b** Schematic for the proposed mechanism of structural recombination. **c** Radar maps comparing the properties of S-LFP, C-LFP, and R-LFMP. **d** Demonstration of the formation of DES. **e** Demonstration of the formation of the precursor. **f** FTIR spectra of original and recycled DES. **g** The separation efficiencies of Li/Fe/Mn/P based on ICP results. The error bars were calculated from three experiments to get the average value. **h** XRD patterns of the precursors. **i** HAADF-STEM image and EDS maps of O, Fe, and Mn. **j** EELS of O *K*-edge, Mn *L*-edge, Fe *L*-edge, and corresponding element contents in the green box area of the HAADF-STEM image.

morphology of the precursor (Supplementary Fig. 5). In addition, in-depth probing by X-ray photoelectron spectroscopy (XPS) shows a uniform concentration of Fe and Mn from the surface to the bulk (Supplementary Fig. 6). The peaks at 709.9 eV and 641.4 eV are attributed to Fe $2p_{3/2}$ and Mn $2p_{3/2}$ respectively, and remained unchanged throughout the sputtering time[39]. High-angle annular dark-field scanning transmission electron microscope (HAADF-STEM) images and energy-dispersive spectroscopy (EDS) elemental maps also show a uniform distribution of Fe and Mn elements in a single particle (Fig. 1i). Electron energy loss spectroscopy (EELS) was used to deter-mine the atomic concentration in a targeted area (green box area in the HAADF-STEM image). The content of Fe is close to that of Mn, sug-gesting a uniform elemental distribution in microregions (Fig. 1j). The same result was found for other particles (Supplementary Figs. 7, 8). Therefore, the above discussion suggests that the upcycling strategy

can achieve simultaneous dissolution and recombination of TMs, thus resulting in a solid solution precursor with uniform distributions of Fe and Mn. DES, serving as both dissolution and precipitation medium, offers significant advantages for processing mixed cathode materials with different crystal structures.

**Structural characterization of R-LFMP**

Due to the above upcycling strategy, the DES system (ChCl-OA) had high leaching efficiencies (>97%) for all elements in black mass, and the subsequent precipitation process of Fe/Mn precursors was also demonstrated, enabling that the molar ratios of Fe/Mn can be precisely controlled in the regenerated materials. A series of R-LFMP materials were synthesized and characterized by the XRD patterns (Supple-mentary Fig. 9) and ICP-OES results (Supplementary Table 2). The Fe/Mn atomic ratios are close to the theoretical values. By comparing the

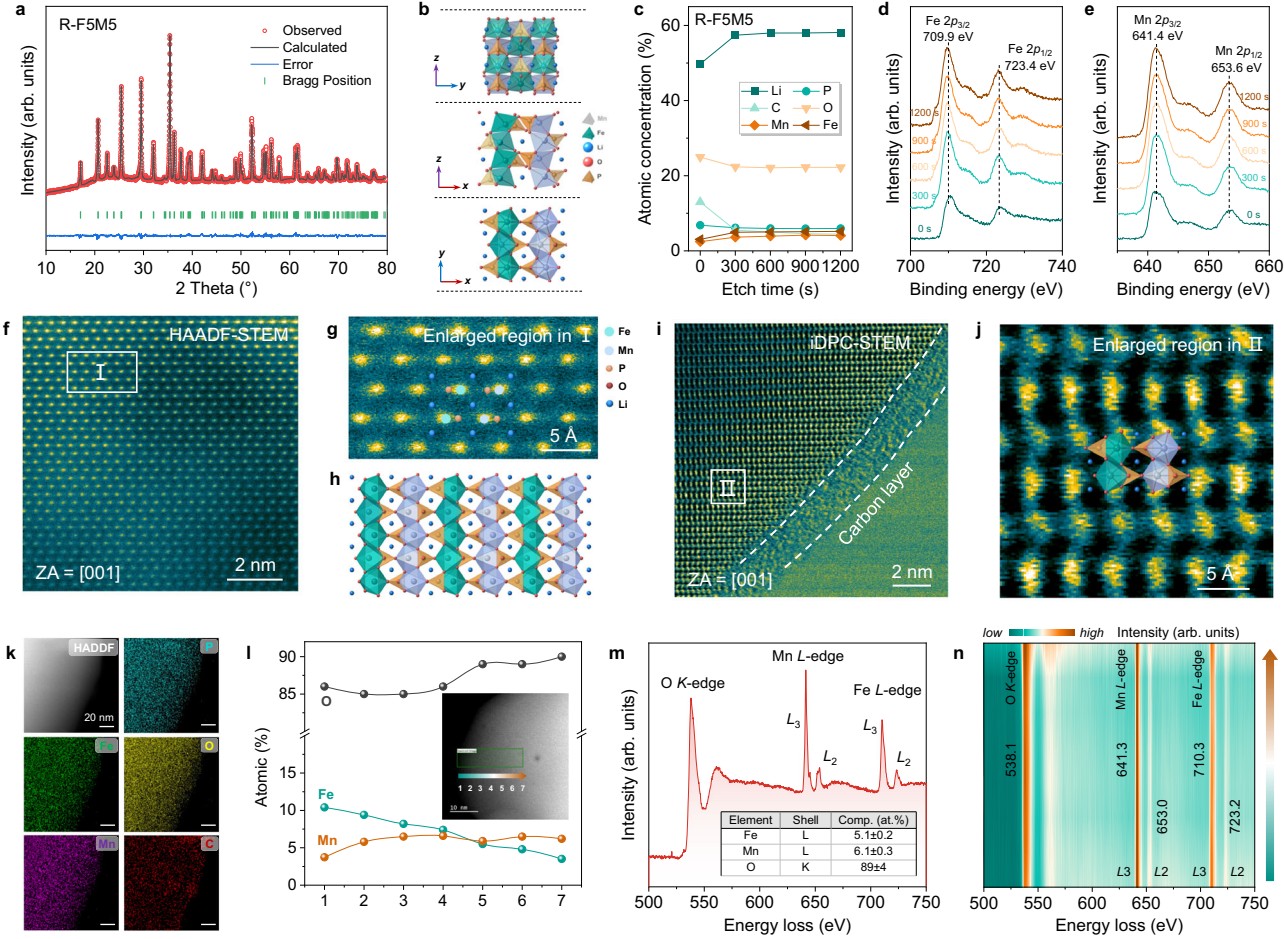

**Fig. 2 | Average structure information, microstructure characterization, and element distributions of R-F5M5. a** XRD pattern and Rietveld refinement results. **b** Schematic of the LFMP crystal structure viewed along [100], [010], and [001]. **c**–**e** Depth etching XPS results. **c** Atomic concentrations, **d** Fe 2*p* XPS, **e** Mn 2*p* XPS. **f** HAADF-STEM image. **g** Enlargement of area I in (**f**). **h** Schematic of the LFMP crystal structure viewed along [001]. **i** iDPC-STEM image. **j** Enlargement of area II in

(**i**). **k** EDS maps. **l** Atomic concentrations along the gradient line. The inset HAADF-STEM image shows the scanning direction of EELS. **m** EELS of the O *K*-edge, Mn *L*-edge, Fe *L*-edge, and the corresponding elemental contents in the green box area. **n** Contour plots of the EELS spectra along the gradient line in the HAADF-STEM image.

performance of the R-LFMP materials (Supplementary Fig. 10), we chose regenerated LiFe$_{0.5}$Mn$_{0.5}$PO$_4$ (R-F5M5) and LiFe$_{0.8}$Mn$_{0.2}$PO$_4$ (R-F8M2) for later research. R-LFMP has an ordered *Pnma* space group (Fig. 2a and Supplementary Fig. 11), as shown by XRD Rietveld refinement, while LFMP has an olivine crystal structure and a larger cell volume than LFP due to Mn having a larger atomic radius than Fe[40] (Fig. 2b and Supplementary Tables 3, 4). Depth XPS profiling shows uniform distributions of Fe and Mn from the surface to the bulk (Fig. 2c and Supplementary Figs. 12, 13), which is determined by the characteristics of the precursors. This upcycling strategy based on DES allows us to control the composition and structure of R-LFMP. Specifically, the peaks at 709.9 eV and 641.4 eV are respectively ascribed to Fe 2*p*$_{3/2}$ and Mn 2*p*$_{3/2}$, suggesting that the valence states of Fe and Mn are +2 in R-F5M5 (Fig. 2d, e). Furthermore, examination of the microstructure of R-LFMP by high-resolution TEM (HRTEM) shows a uniform carbon layer coating on the particle surface with a thickness of 4−5 nm (Supplementary Fig. 14) and HAADF-STEM images showed a regular arrangement of Fe/Mn atoms (Fig. 2f and Supplementary Fig. 15). An enlarged view in Fig. 2g corresponds to the schematic of the LFMP crystal structure viewed along [001] (Fig. 2h). Integrated differential phase-contrast STEM (iDPC-STEM) images show the FeO$_6$/MnO$_6$ octahedron and PO$_4$ tetrahedron at the Å scale (Fig. 2i, j). Also, a carbon coating layer was seen in iDPC-STEM images of the R-F5M5 sample (Supplementary Fig. 16).

EDS elemental maps of an R-F5M5 particle show uniform distributions of Fe, Mn, P, O, and C (Fig. 2k). To determine the specific element contents and distributions, EELS was used. Along the gradient line scan (from point 1 to point 7 in Fig. 2l), the elemental content of Mn is almost unchanged while Fe tends to concentrate on the surface. A similar trend was also seen in another particle (Supplementary Fig. 17). In addition, almost identical concentrations of Fe and Mn (5.1 at% for Fe and 6.1 at% for Mn) were shown in a microregion (the green box area in Fig. 2m). The Mn $L_3$-edge at 641.3 eV and Fe $L_3$-edge at 710.3 eV suggest that the valence states of both metals are +2 in R-F5M5. The O *K*-edge EELS at 583.1 eV shows no pre-edge peak, confirming the low valence states of Fe and Mn[41,42]. From point 1 to point 7 along the gradient line, the Fe *L*-edge and Mn *L*-edge peaks remain unchanged, and no pre-edge peak was detected in the O *K*-edge EELS (Fig. 2n), suggesting that R-F5M5 is a homogeneous solid solution. In addition, the microstructure and element distributions of R-F8M2 were characterized by TEM images and EDS maps (Supplementary Fig. 18). Therefore, R-LFMP materials inherit the advantages of the precursor, that is, the TMs are uniformly distributed, which improves both structural stability and electrical conductivity.

## Electrochemical performance and energy density
The discharge capacities of R-F5M5 and R-F8M2 are respectively 152 and 150 mAh g$^{-1}$ with initial coulombic efficiencies (ICEs) of 94% and

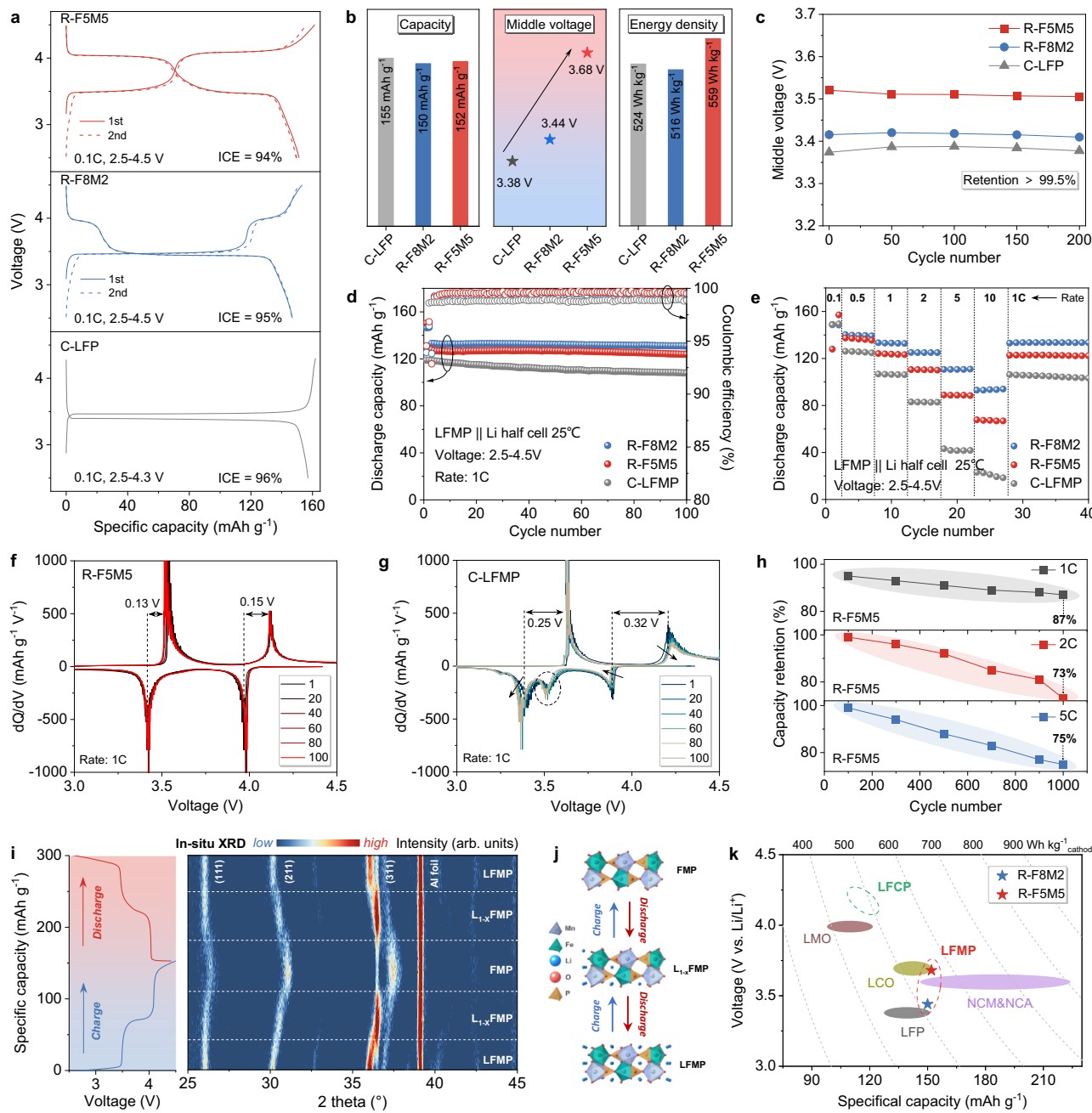

**Fig. 3 | Electrochemical performance and energy densities of LFMP and LFP.**
**a** Initial two charge and discharge curves of R-F5M5, R-F8M2, and C-LFP.
**b** Comparison of the discharge capacities, middle voltages, and energy densities.
**c** Middle voltage retention at 1 C rate after 200 cycles. **d** Cycling performance and **e** rate capabilities of R-F5M5, R-F8M2, and C-LFP. **f**, **g** dQ/dV curves of R-F5M5 and C-LFMP cathodes. **h** Capacity retentions of R-F5M5 at 1, 2, and 5 C. **i** Charge and discharge curves and operando XRD contour map of R-F5M5 in the initial cycle. **j** Schematic of crystal structure changes during charge and discharge. LiFe$_{0.5}$Mn$_{0.5}$PO$_4$, Li$_{1-x}$Fe$_{0.5}$Mn$_{0.5}$PO$_4$ ($1 \geq x \geq 0$) and Fe$_{0.5}$Mn$_{0.5}$PO$_4$ are denoted as LFMP, L$_{1-x}$FMP and FMP, respectively. **k** Comparison of the performance of R-LFMP, R-LFCP, and commercial cathodes.

95% (Fig. 3a). In comparison, a commercial LFP (C-LFP) has the specific capacity of 155 mAh g$^{-1}$ with an ICE of 96%. The middle voltages of R-F8M2 and R-F5M5 respectively increase to 3.44 and 3.68 V (Fig. 3b). As a result, R-F5M5 has an energy density of 559 Wh kg$^{-1}$, higher than that of C-LFP. Even at 1 C rate, >99.5% of the middle voltage is retained after 200 cycles (Fig. 3c). The stable middle voltage is due to the structural advantage, that is the uniform distribution of Fe and Mn in solid solution LFMP. The cycling and rate performance were measured and compared with a commercial LFMP (C-LFMP, the molar ratio of Fe: Mn is 4: 6) cathode (Fig. 3d, e). R-F5M5 and R-F8M2 have discharge capacities of 128 and 134 mAh g$^{-1}$ at 1 C, respectively. When the current

density reached 5 C, capacities of 90 and 110 mAh g$^{-1}$ were still retained for R-F5M5 and R-F8M2, respectively. In contrast, the specific capacity of C-LFMP is 118 mAh g$^{-1}$ at 1 C and decreases to 43 mAh g$^{-1}$ at 5 C.

The cycling stability of LFMP cathodes was also analyzed by dQ/dV curves (Fig. 3f, g). C-LFMP had reduced redox peaks with large polarizations after 100 cycles (0.25 V for Fe$^{+2/+3}$ and 0.32 V for Mn$^{+2/+3}$). An extra peak was observed at 3.5 V during discharge, in agreement with the charge and discharge curves (Supplementary Fig. 19). In comparison, R-F5M5 showed sharp redox peaks with mitigated polarization (0.13 V for Fe$^{+2/+3}$ and 0.15 V for Mn$^{+2/+3}$). Cycling stability was confirmed by CV results in Supplementary Fig. 20. During

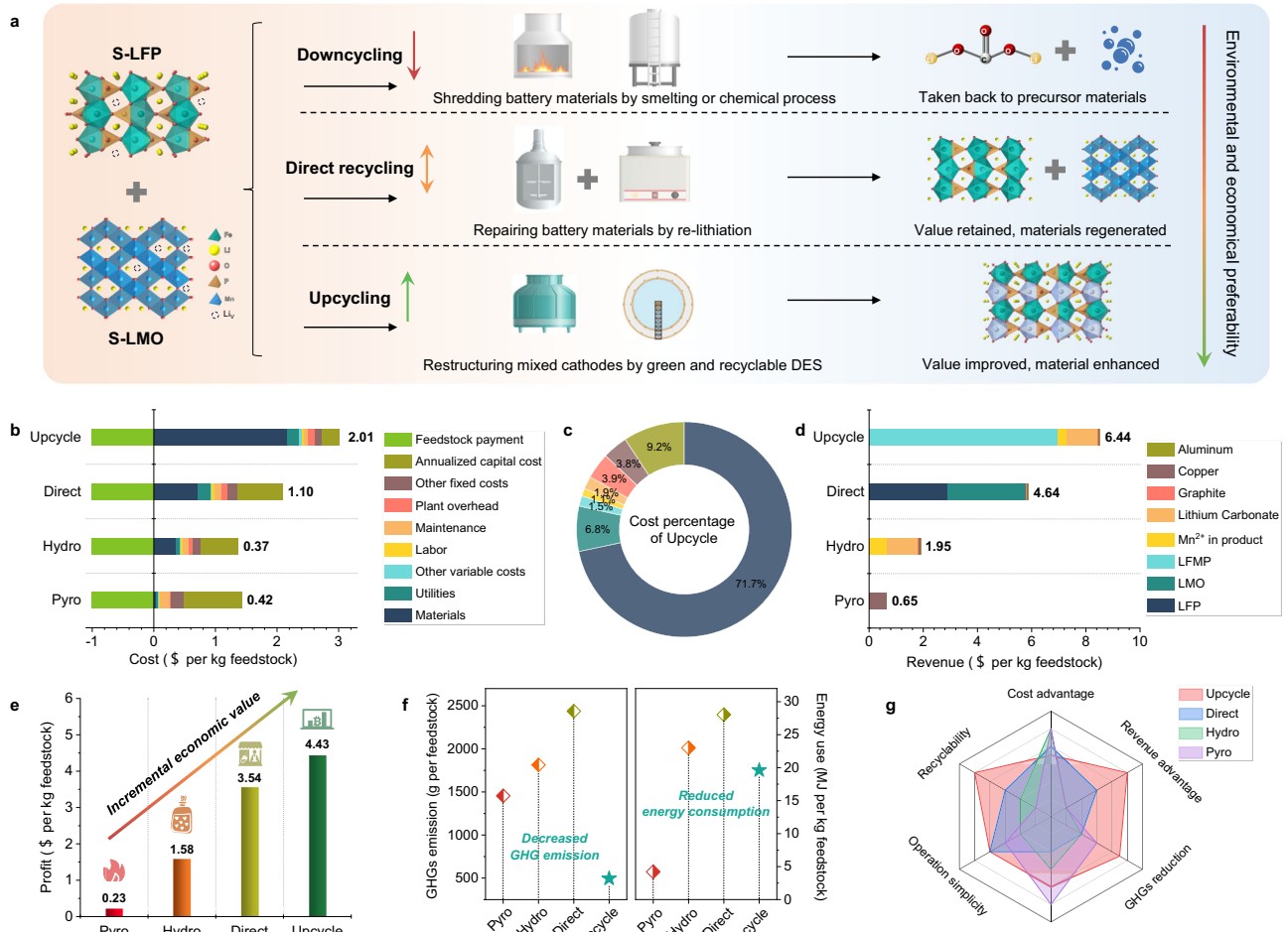

**Fig. 4 | Comparison of different battery recycling technologies and techno-economic analysis. a** Comparison of downcycling, direct recycling, and upcycling processes for S-LFP and S-LMO cathodes. **b**–**f** Techno-economic analysis results of the Pyro, Hydro, Direct, and Upcycle processes. **b** Cost analysis. **c** Cost percentages of the Upcycle process. **d** Revenue analysis. **e** Profit analysis. **f** GHG emission and total energy consumption. **g** Comprehensive comparison of different battery recycling technologies.

long-term cycling measurements, capacity retentions of 87, 73, and 75% were respectively achieved at 1 C, 2 C, and 5 C after 1000 cycles (Fig. 3h and Supplementary Fig. 21). The excellent cycling performance is ascribed to the structural stability. This proposed strategy based on the DES helps generate a uniform distribution of Fe and Mn, resulting in a stable solid solution phase. Moreover, the carbon layer coating improves the electrical conductivity for R-LFMP, so it has a good high-rate capability. To ascertain the superiority of R-LFMP, we synthesized the contrast samples with refined Fe/Mn precursor materials (denoted as P-F8M5 and P-F5M5). The electrochemical performance and detailed discussion were shown in Supplementary Figs. 22–25.

Furthermore, we revealed the mechanism of structural changes during charge and discharge processes by in situ XRD measurements (Fig. 3i). Compared with the behavior of LFP, LFMP transforms through continuous evolution of solid solutions[43,44]. During charging, LFMP undergoes two phase transformations, that is, first from LFMP to an intermediate $Li_{1-x}Fe_{0.5}Mn_{0.5}PO_4$ ($L_{1-x}FMP$) phase and then to $Fe_{0.5}Mn_{0.5}PO_4$ (FMP). The discharge process is inverse, as depicted in Fig. 3j. In addition to LFMP, our proposed strategy is also universally applicable for synthesizing LFCP with higher voltage, and the detailed results are listed in Supplementary Fig. 26 and Table 5. Therefore, R-LFMP not only inherits the advantages of the cycling stability of LFP but also has an increased energy density (Fig. 3k). This is comparable to low-nickel layered oxide cathodes and is expected to be the next-generation phosphate cathode.

## Techno-economic analysis

Figure 4a summarizes different battery recycling methods for mixed spent cathodes, particularly for those containing a low concentration of valuable elements. Downcycling involves traditional pyrometallurgical and hydrometallurgical approaches, which destroy battery cells or materials by high-temperature melting or chemical processes, and suffer from complexity and secondary pollution. The low economic value of the products means that they are unsuitable for recycling S-LFP and S-LMO cathodes. Emerging direct repair/regeneration methods do not break the material structure, but focus on recovering the compositions and structure, so that the regenerated materials can be used for new batteries. These methods retain the value of the recycled products but require delicate processes to separate regenerated materials with different structure. The upcycling approach in our work not only addresses the challenge of mixed spent cathode materials but also achieves value-added upcycling of recycled products. Also, the green and recyclable DES used in our work is also environmentally friendly, thus resulting in a reduced processing cost.

We have made a techno-economic analysis (TEA) of different battery recycling technologies based on the Everbatt 2023 model. Pyrometallurgical recycling (Pyro), hydrometallurgical recycling (Hydro), direct recycling (Direct), and our designed upcycling strategy (Upcycle) are compared for S-LFP and S-LMO cathode materials (Supplementary Figs. 27–30). In the cost analysis, annualized capital cost is a critical factor in all technologies. Material cost is relatively

lower in the Pyro process because the energy required for smelting is produced by components in the battery, such as graphite. In comparison, material cost is a significant contributor to the overall cost in the other processes, such as acid/alkaline reagents in Hydro and lithium salts in Direct and Upcycle. As a result, the recycling costs of Pyro, Hydro, Direct, and Upcycle are respectively 0.42, 0.37, 1.10, and 2.01 $ per kg feedstock (Fig. 4b). The cost breakdown in Fig. 4c shows that material cost accounts for 71.1% of the total. Revenue analysis is directly related to the value of products. Only copper metal is produced in the Pyro process, resulting in low revenue. Lithium carbonate and Mn salts are the main products in the Hydro process. In contrast, the regenerated materials (LFP/LMO/LFMP) are the major contributors to the overall revenue in the Direct and Upcycle processes. In addition, the recycled lithium salt and Mn salts also contribute to the revenue in the Upcycle process. Revenue of 0.65, 1.95, 4.64, and 6.44 $ per kg of feedstock is respectively achieved in the Pyro, Hydro, Direct, and Upcycle processes (Fig. 4d). Our proposed Upcycle process, therefore, has the highest profit, that is 4.43 $ per kg of feedstock, as shown in Fig. 4e.

Energy consumption and environmental impact are also important factors for evaluating battery recycling technologies. As shown in Fig. 4f, the Upcycle process has the lowest greenhouse gas (GHG) emission (495 g per kg feedstock) of all the processes. This can be attributed to the green and recyclable nature of the DES, coupled with the high recycling efficiency of all elements. In addition, the total energy consumption of the Upcycle process is 19.6 MJ per kg of feedstock, which is lower than that of Hydro and Direct processes. The four battery recycling technologies have been comprehensively compared, as summarized in Fig. 4g. The Pyro process is unsuitable for recycling mixed cathodes due to its poor economic value and recyclability, especially for S-LFP and S-LMO. The Hydro process requires strong acid reagents and complicated steps, resulting in high material consumption and labor requirements. The Direct process is an emerging technology aimed at restoring the cathodes to their original state[32,45]. It has practical problems in processing mixed cathodes in the industry due to structural incompatibility. In contrast, our proposed Upcycle process has significant advantages in revenue because the regenerated LFMP is expected to be a next-generation phosphate cathode, and the use of recyclable DES accounts for the low GHG emission and acceptable energy consumption. The Upcycle process, therefore, presents a feasible solution for the sustainable recycling of mixed spent cathode materials, producing a regenerated material with enhanced performance.

## Discussion

We have reported a sustainable upcycling approach for recycling mixed spent cathode materials, which were converted into a high-voltage polyanionic cathode with improved energy density. The use of recyclable DES aligns with the principle of a circular economy, ensuring that all the elements were efficiently recycled. R-LFMP is a solid solution with a uniform distribution of the Fe/Mn elements, which both enhances the structural stability and improves its electrical conductivity. A detailed TEA shows that this upcycling strategy has potential advantages in economic and environmental benefits compared with the other recycling technologies. Our approach provides an alternative and simple route for the recovery of mixed spent cathode materials, leading to next-generation phosphate cathodes for practical LIBs.

Considering that the potential feasibility for practical application is crucial for evaluating a battery recycling process, we carried out a scalable experiment demonstration (Supplementary Fig. 31). As expected, the experimental phenomenon was consistent with the previous processes, and the R-LFMP samples synthesized from the scale-up experiments showed satisfying properties

(Supplementary Fig. 32), demonstrating that the upcycling approach was able to expand at a gram-grade scale. Furthermore, we have expanded the scope of practical applications of the upcycling strategy, that is, a potential recycling approach for LFMP was proposed and verified by experiments (Supplementary Fig. 33). A sustainable recycling system, involving the upcycling strategy in this work and potential LFMP recycling in the future, is summarized in Supplementary Fig. 34. LFMP, regarded as the structural upgrade product of LFP, has attracted much attention in both academic and industrial fields in recent years. The DES system in this work was also applicable to process LFMP, suggesting that the proposed upcycling strategy can achieve a closed-loop recycling between the mixed spent cathode chemistries and the next generation cathode materials.

With the rapid development of electric vehicles in recent years, different types of LIBs involving olivine LFP, layered oxide LCO/NCM, and spinel LMO will face a wave of retirement. In contrast, LFP and LMO batteries with relatively lower manufacturing costs still account for a considerable market share in areas of large-scale energy storage station and electric vehicles. Given the limited energy density of LFP on its own, we believe that upcycling mixed S-LFP and Mn-rich materials to a high-voltage polyanionic cathode material is a promising approach. It serves as a bridge for the recycling of degraded cathode materials and the regeneration of next-generation cathode materials. Many components (such as LFP, LMO, and Graphite) in batteries have relatively lower economic values, and most current recycling strategies are unsuitable. How to upcycle these materials and enhance the value of recycled products is a crucial issue. LFMP reported here is one example of improving the energy density, and there are more directions in the future, such as improving the low-temperature performance of recycled materials, and exploring value-added applications of recycled products in other fields. These potential directions are worth further exploration and sustainable upcycling technologies will play a significant role in generating greater environmental and economic benefits.

## Methods
### Materials and chemicals
Spent LFP and LMO black mass were all obtained from a battery company in China. Spent LCO cathode powder was separated from a mobile phone battery. Chemicals involve choline chloride ($C_5H_{14}ClNO$, ChCl, 98%), oxalic acid ($H_2C_2O_4$, OA, 99%), lithium hydroxide (LiOH, 99%), ammonium dihydrogen phosphate ($NH_4H_2PO_4$, 98%), $FeC_2O_4$ (99%), $MnC_2O_4 \cdot 2H_2O$ (99%), ethanol (98%), polyvinylidene fluoride (PVDF, 99%), $N$-methyl-pyrrolidone (NMP, 99%), acetylene black (AB, 98%), and carbon nanotubes (CNT, >90%, OD: 8–15 nm, length: 30–50 μm). All of the chemicals are analytical grade and purchased from Macklin.

### Preparation of DES
ChCl and OA were mixed in a molar ratio of 1: 1 and heated at 80 °C under stirring to form a transparent liquid. The obtained DES was then used for leaching the mixed black mass.

### Synthesis of the $MeC_2O_4 \cdot 2H_2O$ precursor
S-LFP and S-LMO black mass with a specific Fe/Mn molar ratio were added to DES and the mixture was then heated at 110 °C under continuous stirring for 6 h. After cooling to room temperature, an equal volume of deionized water was added to form a homogeneous solution for standing overnight. After filtration and washing, a precipitate of (Fe, Mn)$C_2O_4 \cdot 2H_2O$ was dried at 60 °C to remove the excess water. The DES was recycled by heating the filtered liquid to remove water and reused for the next cycle. Notably, the Fe/Mn molar ratios in the precursors can be precisely controlled in black mass due to the high leaching efficiencies for all elements, which has been demonstrated in

Results section. The synthesis of $(Fe, Co)C_2O_4\cdot 2H_2O$ precursor is the same as for $(Fe, Mn)C_2O_4\cdot 2H_2O$ except for the spent LCO raw material.

## Synthesis of LFMP and LFCP

$(Fe, Mn)C_2O_4\cdot 2H_2O$, $NH_4H_2PO_4$, and LiOH in a molar of 1: 1: 1.05 were mixed in a planetary ball mill for 6 h. An appropriate amount of ethanol was used as the milling medium. After drying at 90 °C under vacuum, the mixture was then heated at 5 °C min$^{-1}$ to 350 °C where it was sintered for 6 h; after which the mixture was heated to 650 °C at the same heating rate and sintered for 10 h. The LFMP powder was synthesized after natural cooling to room temperature. The experimental conditions were the same as for the R-LFMP with different molar ratios of Fe/Mn. The contrast samples (P-F8M5, P-F5M5) were synthesized using the same experimental conditions except for the commercial $FeC_2O_4$ and $MnC_2O_4\cdot 2H_2O$ precursors. The synthesis of LFCP is the same as for LFMP except for the $(Fe, Co)C_2O_4\cdot 2H_2O$ precursor.

## Reuse of the DES and recycling of the lithium salt

To regenerate the DES, the filtered liquid was heated to evaporate off the excess water and reused for the next cycle. After being used several times, the oxalate content can be adjusted through extra addition for reforming the DES, based on how many moles were consumed in the precipitate. To recycle lithium and phosphate, the pH of the filtered liquid containing high concentrations of lithium and phosphate was first changed to ~10. After filtration to remove the residual Fe/Mn precipitate, the filtered liquid was heated to remove most of the water. Lithium was precipitated in the form of lithium phosphate without needing any precipitating agent.

## Gram-grade scale experimental demonstration

About 56 g ChCl and 36 g OA were mixed in a molar ratio of 1: 1 to form the DES, which was used for leaching the black mass. Then 1.58 g S-LFP and 0.90 g LMO (Fe: Mn = 1: 1 in a molar ratio) were added into the DES and the mixture was then heated at 110 °C under continuous stirring for 6 h. The subsequent experimental procedures were the same as those in the Methods section.

## Characterizations

XRD patterns were collected using a Bruker D8 Advance (operating tube voltage of 40 kV, tube current at 30 mA, Cu Kα λ = 1.5406 Å, Germany). The element valances were determined by XPS spectroscopy (Thermo Fisher Scientific K-Alpha, USA). ICP-OES (Agilent ICPOES730, USA) was used to determine element contents. FTIR spectra were recorded using a Nexus 670 FTIR spectrometer over the range of 500–4000 cm$^{-1}$. The morphologies and microstructure were characterized by field-emission SEM (Zeiss, S-3500N, Japan) and cold-field-emission spherical aberration corrected TEM (Thermo Fisher Scientific, Spectra 300, USA). EELS was used to analyze the surface elemental valence states and concentration distributions.

## Battery assembly and electrochemical test

The active materials, AB, CNT, and PVDF binder, were mixed in weight ratios of 80: 5: 5: 10. The mixture was dispersed in an NMP solution to form a slurry, after which it was coated on an Al foil. The mass loading was controlled to 4–5 mg cm$^{-2}$. Coin cells (CR2032) were assembled to test the electrochemical performance of the LFMP and LFP samples. Li metal chips, Celgard 2325, and 1.0 M LiPF$_6$ in ethylene carbonate/dimethyl carbonate/diethyl carbonate (EC: DMC: DEC = 1: 1: 1 in volume) were used as the counter electrode, separator, and electrolyte, respectively. The assembled cells were tested on a NEWARE battery test system (CT-4008T-5V 20 mA, Shenzhen, China). The cycling performance was tested in the voltage range of 2.5–4.5 V (versus Li/Li$^+$) and the rate capability test used current densities from 0.1–10 C (1 C = 170 mAh g$^{-1}$). The CV and EIS tests were conducted on an electrochemical workstation (BioLogic SP-150e).

## TEA and LCA analysis

We made a techno-economic analysis and life-cycle analysis of the Pyro, Hydro, Direct, and Upcycle recycling processes based on the EverBatt 2023 model, which was developed by Argonne National Laboratory. We input 10,000 tonnes of spent LFP and 10,000 tonnes of spent LMO end-of-life batteries for preprocessing (Supplementary Table 6) and the separated black mass was used for the different recycling processes. The detailed cost of raw materials and revenue of products are listed in Supplementary Tables 7–11. The GHG emissions and total energy consumption are listed in Supplementary Table 12.

## Reporting summary

Further information on research design is available in the Nature Portfolio Reporting Summary linked to this article.

## Data availability

The datasets generated during and/or analysed during the current study are available from the corresponding author on reasonable request. Source data are provided with this paper.

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

## Acknowledgements

G.Z. appreciates the support from the Joint Funds of the National Natural Science Foundation of China (U21A20174), Guangdong Innovative and Entrepreneurial Research Team Program (2021ZT09L197), Shenzhen Science and Technology Program (KQTD20210811090112002), the Startup Funds, and Interdisciplinary Research and Innovation Fund of Tsinghua Shenzhen International Graduate School. G.Z. and J.W. appreciate support from Qinhe Energy Conservation and Environmental Protection Group Co., Ltd. (No. QHHB-20210405). Z.L. acknowledges financial support from the startup funds of Shanghai Jiao Tong University. J.W. appreciates support from the Startup Fund for Young Faculty at SJTU (23×010502206), National Natural Science Youth Fund (52302285). This work made use of the TEM facilities at the Institute of Materials Research, Tsinghua Shenzhen International Graduate School (Tsinghua SIGS).

## Author contributions

G.J., J.W., G.Z., and H.-M.C. conceived the project. G.J and D.T. fabricated the samples, performed electrochemical measurements, and analyzed the data with the assistance of H.J., J.M., Z.Z., S.L. Z.L., G.Z., and H.-M.C. supervised the research and revised the manuscript. All authors discussed and contributed to the results. G.J. wrote the manuscript with comments and revisions from all the authors.

## Competing interests

The authors declare no competing interests.
