## [Peer Review File · Nature Communications]

Sustainable upcycling of mixed spent cathodes to a high-voltage polyanionic cathode materialREVIEWER COMMENTS

Reviewer #1 (Remarks to the Author):

In this study, the authors have presented an innovative approach to battery recycling through structural design and transition metal replacement using deep eutectic solvents. While the approach is intriguing, I believe that further enhancements to the manuscript are necessary in order to more robustly substantiate the authors' assertions through additional experimentation and discussion.

Specific comments:

1. In Figure 1, there should be a quantitative measurement of the dissolution of S-LFP and S-LMO in the deep eutectic solvent (DES). Such quantitative data is essential to support the author's claims and to provide a clearer understanding of the LFMP preparation process, including the composition of elements and the concentration in the leaching solution of the black mass with DES. While the authors have presented indirect evidence using SEM, XRD, and XPS, it would be more comprehensive to include scientific evidence regarding the elements in the leaching solution.
2. The discussion regarding the control of the Fe/Mn molar ratio in the upcycling strategy is somewhat ambiguous. It would be beneficial to clarify how this ratio is controlled and to delve deeper into this aspect in the manuscript.
3. To ascertain the superiority of the prepared samples, it is crucial to analyze the electrochemical performance of these samples in comparison to LFMP synthesized with highly purified and well-defined raw materials. I recommend including results from LFMP prepared with such refined raw materials and expanding upon the discussion in this context.
4. A more comprehensive discussion on the scale-up strategy for practical application is important to demonstrate the feasibility and effectiveness of the proposed battery recycling process. Please consider adding an in-depth discussion on the practical applications of this approach.

Reviewer #2 (Remarks to the Author):

Ji et al. report an upcycling strategy for mixed degraded cathode materials and convert them into a promising polyanionic cathode with an increased working potential. Currently, most battery recycling reports aim at a single component, and tackling the multi-component cathodes is a challenge especially for phosphate cathodes and Mn-rich spinel cathodes. They have relatively lower economic value, thus needing sustainable recycling methods. The authors propose an efficient and interesting method to recover mixed cathodes by using a green and recyclable DES. The authors also provide an estimation based on a techno-economic analysis, which is helpful for the analysis if such technologies can be applied on large scale.

I would recommend the manuscript for publication in Nature Communications after having addressed subsequent issues:

1. The DES (ChCl-OA system) used in this work was green and recyclable, as shown in Figs. 1d-f. How is the DES recycled in the process considering oxalates were actually consumed. More discussions are needed in the experimental method part.
2. A uniform carbon layer coating can be seen in STEM images (Fig. 2i) and EDS maps (Fig. 2k). Are additional carbon sources added during the recovery process? A brief discussion

along these lines is required.

3. The choice of Fe and Mn mole ratio seems to be an important factor, which is related to the performance. Please provide a detailed explanation about how to exactly control the ratios in the R-LFMP materials.

4. For the TEA analysis in Fig. 4, the upcycle process has a higher cost compared with Pyro, Hydro, Direct processes. What could be the reason?

Reviewer #3 (Remarks to the Author):

This paper explains a new recycling route for the cathode of a lithium ion battery. The novelty is in dissolving 2 types of cathode material (lithium iron phosphate), termed LFP, and lithium manganese oxide (LiMn_2O_4), termed LMO, to produce lithium iron manganese phosphate, termed LFMP. The use of LFMP as a promising cathode material has been known to the literature for about 1 decade as noted by the authors. The novelty of the paper is the fact that the authors have come up with a processing route that uses a so-called deep eutectic solvent (DES) to manufacture LFMP. The mixture of many chemicals used to react with the 2 cathode materials are given in the methods section without any justification. For example, carbon nanotubes are used without explanation.

The authors show theoretically that the processing route is environmentally friendly and is cost effective.

The authors have manufactured and characterised two recycled forms of LFMP which they describe as R-F5M5 and R-F8M2. They do not explain the differences in manufacturing route for each. Lots of detail on the chemical structure of R-F5M5 is given in Fig. 2, but there is almost no information given on R-F8M2. The electrochemical performance of the two alloys are compared in Fig. 3. The paper is imbalanced in this regard.

The battery properties of LFMP are an improvement upon those of LFP and LMO, but are not outstanding.

The paper is highly detailed in its electrochemical results, and it is not clear to the reviewer that this will appeal to the general audience.

There is no mention on how the authors propose to recycle the LFMP.

The English is poor in some places. For example, the authors frequently use the word 'even' to mean 'uniform'.

Point-by-point response to the reviewers' comments

Reviewer #1 (Revision marked with red in the revised MS and SI)

In this study, the authors have presented an innovative approach to battery recycling through structural design and transition metal replacement using deep eutectic solvents. While the approach is intriguing, I believe that further enhancements to the manuscript are necessary in order to more robustly substantiate the authors' assertions through additional experimentation and discussion.

Response: Thanks for your suggestions on this manuscript, which are helpful for revision of this manuscript. We have added necessary experiments to determine the dissolution behavior of the elements in the deep eutectic solvent (DES). The composition and concentration of the black mass were further revealed by ICP and other characterizations. In addition, we supplemented the data of two comparison samples, which were synthesized with highly purified and well-defined raw materials. Finally, we also conducted a scalable experiment to verify the feasibility of this recycling process and added an in-depth discussion on the practical applications. All the changes were highlighted with red in the revised MS and SI. Below are the point-to-point responses to your comments.

Specific comments:

1. In Figure 1, there should be a quantitative measurement of the dissolution of S-LFP and S-LMO in the deep eutectic solvent (DES). Such quantitative data is essential to support the author's claims and to provide a clearer understanding of the LFMP preparation process, including the composition of elements and the concentration in the leaching solution of the black mass with DES. While the authors have presented indirect evidence using SEM, XRD, and XPS, it would be more comprehensive to include scientific evidence regarding the elements in the leaching solution.

Response: Thanks for your comments. We have added quantitative measurements to determine the dissolution of S-LFP and S-LMO in the DES. The ICP results (**Table R1** and **Fig. R1**) showed that the leaching rates of Li and P were up to 96.7% and 97.0%, respectively. Lithium and phosphorus remained respectively as Li^+ and PO_4^{3-} in the DES, which were then recycled as Li_3PO_4 from the filtrate, as confirmed by the XRD pattern (**Supplementary Fig. 3**). After the leaching reaction, iron and manganese remained respectively as $[\text{FeCl}_4]^{2-}$ and $[\text{MnCl}_4]^{2-}$ in the DES, which were precipitated as $(\text{Fe, Mn})\text{C}_2\text{O}_4 \cdot 2\text{H}_2\text{O}$ in the following separation processes. Notably, the DES has greater viscosity than aqueous solutions with majority >100 cP at ambient temperature, making it difficult to directly measure the leaching rates in the DES. Hence, the leaching rates were calculated based the filtrate after removing $(\text{Fe, Mn})\text{C}_2\text{O}_4 \cdot 2\text{H}_2\text{O}$ precipitation. According to the high leaching efficiencies of Li and P ($\geq 97\%$), we inferred that the Fe and Mn can also be leached with the fast reaction kinetics. This is because the structures of LFP and LMO cathodes were broken down and rearranged in

the DES system. Therefore, 89.6% of the Fe and 82.2% of the Mn were precipitated as the oxalates after adding water for the separation processes.

Table R1 | The leaching rates of Li/P/Fe/Mn based on ICP results.

	Li	P	Fe	Mn
Filtrate-1	0.991	0.957	0.113	0.206
Filtrate-2	0.952	0.973	0.074	0.141
Filtrate-3	0.959	0.981	0.126	0.185

Note: 1 mmol S-LFP and 0.5 mmol S-LMO were used to calculate the leaching rate. Three experiments were used to get the average value. The leaching rates were calculated based the filtrate after removing $(\text{Fe, Mn})\text{C}_2\text{O}_4 \cdot 2\text{H}_2\text{O}$ precipitation.

Fig. R1 | The separation rate of Li/P/Fe/Mn based on ICP results.

We attempted to use FTIR spectra to characterize the coordination environment and composition of the dissolved elements in the DES (**Fig. R2**). Unfortunately, these infrared peaks of these elements were difficult to detect because the DES is an organic solvent, and the peaks of its own organic functional groups were too strong, thus masking the appearance of other peaks. Combined with other characterizations and literature support, we can give a qualitative analysis of the dissolved elements in the DES.

Fig. R2 | FTIR spectra of pristine DES and the black mass dissolved in the DES at different times.

For a clearer and deeper understanding of the upcycling strategy, especially for the dissolution and rearrangement of S-LFP and S-LMO cathodes in the DES. We have added a schematic to demonstrate the core ideas and details of this work (**Fig. R3**). S-LFP (space group: *Pnma*, olivine) and S-LMO (space group: *Fd-3m*, spinel) have different crystal structures, making them incompatible with the direct synthesis of polyanion-type LFMP. Hence, the breakdown and recombination of the mixed materials are necessary in the upcycling process. **The proposed dissolution mechanism is elucidated as follows:** hydrogen in the OA molecules attacked the olivine structure of LFP and the spinel structure of LMO by destroying the Li-O and Me-O (Me = Fe/Mn) bonds. Then the $(C_2O_4)^{2-}$ or ChCl reduced $Me^{3+/4+}$ to Me^{2+} , further promoting the leaching of Me and resulting in the collapse of the crystal structure. Li and Me dissolved in the DES combined with chlorine in ChCl to form stable $[LiCl_2]^-$ and $[MeCl_4]^{2-}$ complexes^{1,2}. Furthermore, the introduce of water as a diluent decreased the inherently high viscosity of the DES, thus transforming the $[MeCl_4]^{2-}$ to $MeC_2O_4 \cdot 2H_2O$ precipitation^{3,4}. The isolated Li^+ and PO_4^{3-} were recycled as a lithium salt and the DES

was reused for the next loops.

We have replaced the original **Fig. 1b** by the following schematic and added the corresponding description on pages 6-7 in the revised MS, which will be useful for understanding the dissolution and rearrangement of S-LFP and S-LMO cathodes in the DES.

Fig. R3 | Schematic for the upcycling strategy.

2. The discussion regarding the control of the Fe/Mn molar ratio in the upcycling strategy is somewhat ambiguous. It would be beneficial to clarify how this ratio is controlled and to delve deeper into this aspect in the manuscript.

Response: Thanks for your suggestions. The control of the Fe/Mn molar ratio is crucial for the regenerated materials in the upcycling processes. As for this important concern, we first re-explained it in the Methods part and the main body part, respectively, in order to show a clear description of this study for the readers.

On pages 9-10 in the revised MS:

“Due to the above upcycling strategy, the DES system (ChCl-OA) had high

leaching efficiencies (> 97%) for all elements in black mass, and the subsequent precipitation process of Fe/Mn precursors was also demonstrated, enabling that the molar ratios of Fe/Mn can be precisely controlled in the regenerated materials...

On pages 19 and 20 in the revised MS (Methods section):

“S-LFP and S-LMO black mass with a specific Fe/Mn molar ratio were added to the DES and the mixture was then heated at 110 °C under continuous stirring for 6 h...”

“Notably, the Fe/Mn molar ratios in the precursors can be precisely controlled in black mass due to the high leaching efficiencies for all elements, which has been demonstrated in Results part...”

“The experimental conditions were the same as for the LFMP materials with different molar ratios of Fe/Mn...”

And, we further illustrate the control of the Fe/Mn molar ratio by combining the relevant experimental data, that is, if the Fe/Mn molar ratio is accurate in the black mass, the ratio is ideal in the precursor and LFMP materials. This is because the DES system used for the upcycling strategy has high leaching efficiencies for all the elements in S-LFP and S-LMO cathodes (**Fig. R1**). In addition, the subsequent precipitation process of Fe/Mn has displayed in **Fig. R4**. Our strategy also shows a high separation efficiency of all elements. 89.6% of the Fe and 82.2% of the Mn were precipitated in the solid solution precursor. While 96.7% of the Li and 97.0% of the P remained in the filtrate, which were then recycled as a lithium salt (Li₃PO₄).

Fig. R4 | d Demonstration of the formation of DES. **e** Demonstration of the formation of the precursor. **f** FT-IR spectra of original and recycled DES. **g** The separation efficiencies of Li/Fe/Mn/P based on ICP results.

Based on above discussion, the Fe/Mn molar ratio in the upcycling strategy is controllable. Now we believe that the revised MS will provide a clear description on the main results of this study.

3. To ascertain the superiority of the prepared samples, it is crucial to analyze the electrochemical performance of these samples in comparison to LFMP synthesized with highly purified and well-defined raw materials. I recommend including results from LFMP prepared with such refined raw materials and expanding upon the discussion in this context.

Response: Thanks for your comments. We have synthesized the comparative samples with highly purified raw materials (Chemicals involve FeC_2O_4 , $\text{MnC}_2\text{O}_4 \cdot 2\text{H}_2\text{O}$, which are analytical grade and purchased from Macklin), as shown in **Fig. R5**. All the experimental conditions were the same as for the R-LFMP materials except for the commercial Fe/Mn precursors. Here we take pristine $\text{LiFe}_{0.5}\text{Mn}_{0.5}\text{PO}_4$ (P-F5M5) as a comparison sample for discussion. As shown in **Fig. R6**, P-F5M5 had an initial discharge capacity of 130 mAh g^{-1} with a low initial Coulombic efficiency (ICE) of 68%. As contrast, the initial discharge capacity of R-F5M5 was 152 mAh g^{-1} with ICE of 94%, indicating the superior performance and cyclic reversibility. Regarding the rate performance (0.1-0.5-1-2-5-10C), the discharge capacities of 128, 110 and 90 mAh g^{-1} were still retained at 1, 2, and 5C rates for R-F5M5. In contrast, P-F5M5 only had

discharge capacities of 100, 86, and 64 mAh g⁻¹ under the same rates.

Fig. R5 | XRD patterns of P-LFMP samples.

Fig. R6 | Electrochemical performance of P-F5M5. a, b Initial two charge and discharge curves at 0.1C. **c, d** Rate capabilities and cycling performance at 1C rate.

To further reveal the reasons for the differences in electrochemical performance, we compared and analyzed the dQ/dV curves and in-situ EIS results. R-F5M5 showed sharp redox peaks with mitigated polarization (0.13 V for Fe^{+2/+3} and 0.15 V for Mn^{+2/+3}). In comparison, P-F5M5 had larger polarizations after 100 cycles (0.13 V for Fe^{+2/+3} and 0.20 V for Mn^{+2/+3}). An extra peak was observed at 3.6 V during discharge, in agreement with the charge and discharge curves. The same phenomenon was seen in the C-LFMP

samples (**Fig. 3g**), ascribed with the structural instability.

Fig. R7 | dQ/dV curves from 1st to 100th cycle at 1C rate. **a** R-F5M5, **b** P-F5M5.

In addition, in-situ EIS measurements were carried out to clarify the change of internal resistance (**Fig. R8**). The semicircles at a high frequency are ascribed to the charge transfer contribution from the cathode interfaces. During charging, the charge transfer impedance (R_{ct}) decreased gradually and reached a minimum at 100% state of charge (4.5 V). R-F5M5 had a smaller R_{ct} value compared with P-F5M5, indicating that a better electronic and ionic conductivity of the R-F5M5 cathode during delithiation. This also explains why the capacity release and cycling stability of R-F5M5 samples are better than that of P-F5M5. During discharging, the R_{ct} increased slightly and then remained at a steady state. The two samples showed similar results. **We have added the above electrochemical performance results and discussion in the revised MS (on page 14) and SI (on pages 15-18).**

Fig. R8 | Impedance spectra collected during the first cycle. a R-F5M5, b P-F5M5. c R_{ct} value change trends.

4. A more comprehensive discussion on the scale-up strategy for practical application is important to demonstrate the feasibility and effectiveness of the proposed battery recycling process. Please consider adding an in-depth discussion on the practical applications of this approach.

Response: Thanks for your helpful comments. The potential feasibility for practical application is crucial for evaluating a battery recycling process. Also, cost and environmental issues are also important considerations in developing a sustainable recycling system. **We first carried out a scalable experiment in the lab (Ten times of the processes in the MS), as depicted in Fig. R9.** Specifically, 56 g choline chloride (ChCl) and 36 g oxalic acid (OA) were mixed in a molar ratio of 1: 1. Then the mixture was heated at 80 °C to form a transparent DES, which was used for leaching the black mass.

After which, 1.58 g S-LFP and 0.90 g LMO (Fe: Mn = 1: 1 in a molar ratio) were added into the DES and the mixture was then heated at 110 °C under continuous stirring for 6 h. The subsequent experimental procedures were exactly the same as that in the Methods section. As expected, the experimental phenomenon was consistent with the previous processes. Therefore, the upcycling approach is able to expanding at a gram-grade scale. Unfortunately, it is difficult for us to carry out a larger demonstration at the laboratory level, but we believe that this upcycling approach has the potential for practical large-scale applications, and we will continue to expand in this direction in the future.

Fig. R9 | Pictures for a scalable experiment in the lab.

To verify the feasibility of the whole upcycling processes on a large-scale, we further synthesized the R-F5M5-scale sample by using the precursor obtained from a scalable experiment, as shown in Fig. R10. XRD patterns show good phase crystallinity for both (Fe, Mn) $C_2O_4 \cdot 2H_2O$ precursor and R-F5M5-scale sample. In electrochemical properties tests, it delivered a discharge capacity of 130 mAh g^{-1} at 1C rate, which was a very

satisfying result.

Fig. R10 | **a** XRD patterns for the precursor and R-F5M5 synthesized by the scale-up experiments. **b** Electrochemical performance of R-F5M5-scale sample.

In addition, we have expanded the scope of practical applications of the upcycling strategy, that is, **a potential recycling approach for LFMP was proposed and verified by experiments**. LFMP has the same olivine type crystal structure as LFP and can be dissolved by the DES (ChCl-OA). As shown in **Fig. R11**, here we used a commercial LFMP (C-LFMP) and R-LFMP as the raw materials, respectively. The same target product was formed, that is, the Fe/Mn oxalate precursor, suggesting that **the upcycling approach in this work is also applicative to process LFMP cathodes**. Based on above discussion, **a sustainable recycling system, involving the upcycling strategy in this work and potential LFMP recycling in the future**, is summarized in **Fig. R12**. LFMP is a promising cathode material with an increased voltage and energy density. Especially in recent years, many battery manufacturers began to layout LFMP batteries. We believe that LFMP will usher in explosive growth in the next few years, so it can be inferred that sustainable recycling of LFMP batteries will also be a hot topic.

Fig. R11 | a Picture for C-LFMP and R-LFMP dissolved in the DES. b XRD patterns for the obtained precursors.

Fig. R12 | Schematic for the upcycling strategy in this work and the potential LFMP recycling in the future.

Finally, we have added an in-depth discussion on the practical applications of this approach, as shown in revised MS and SI.

On page 19 in the revised MS (Discussion section):

“Considering that the potential feasibility for practical application is crucial for evaluating a battery recycling process, we carried out a small scale-up experiments demonstration (Supplementary Fig. 31). As expected, the experimental phenomenon was consistent with the previous processes, and the R-LFMP samples synthesized from the scale-up experiments showed satisfying properties (Supplementary Fig. 32), demonstrating that the upcycling approach was able to expanding at a gram-grade

scale. Furthermore, we have expanded the scope of practical applications of the upcycling strategy, that is, a potential recycling approach for LFMP was proposed and verified by experiments (Supplementary Fig. 33). A sustainable recycling system, involving the upcycling strategy in this work and potential LFMP recycling in the future, is summarized in Supplementary Fig. 34. LFMP, regarded as the structural upgrade product of LFP, has attracted much attention in both academic and industrial fields in recent years. The DES system in this work was also applicable to process LFMP, suggesting that the proposed upcycling strategy can achieve a closed-loop recycling between the mixed spent cathode chemistries and the next-generation batteries.”

Reviewer #2 (Revision marked with red in the revised MS and SI)

Ji et al. report an upcycling strategy for mixed degraded cathode materials and convert them into a promising polyanionic cathode with an increased working potential. Currently, most battery recycling reports aim at a single component, and tackling the multi-component cathodes is a challenge especially for phosphate cathodes and Mn-rich spinel cathodes. They have relatively lower economic value, thus needing sustainable recycling methods. The authors propose an efficient and interesting method to recover mixed cathodes by using a green and recyclable DES. The authors also provide an estimation based on a techno-economic analysis, which is helpful for the analysis if such technologies can be applied on large scale. I would recommend the manuscript for publication in Nature Communications after having addressed subsequent issues:

Response: Thanks for your comments on this manuscript, which are useful for us to revise the paper. We have done a full investigation and added some experiments to address your concerns. **All the changes were highlighted with red in the revised manuscript (MS) and supplementary information (SI).** Below are the point-to-point responses to your comments.

1. The DES (ChCl-OA system) used in this work was green and recyclable, as shown in Figs. 1d-f. How is the DES recycled in the process considering oxalates were actually consumed. More discussions are needed in the experimental method part.

Response: Thanks for your comments. The FTIR results in **Fig. 1f** showed that the recycled DES had the same infrared peaks compared with the original DES after the first use, suggesting the remained properties of the DES. This is because the liquid-solid ratio of the reaction is roughly 35 and the oxalate is very abundant. Considering that the oxalates were actually consumed after using for several times, the extra oxalate was needed to adjust appropriate proportion of ChCl and OA, based on how many moles were removed in the precipitate. **We have added a detailed explanation about the reuse of the DES in the Method section, as shown on page 22 in the revised MS:**

“Reuse of the DES and recycling of the lithium salt. To regenerate the DES, the filtered liquid was heated to evaporate off the excess water and reused for the next cycle. After being used several times, the oxalate content can be adjusted through extra addition for reforming the DES, based on how many moles were consumed in the precipitate...”

2. A uniform carbon layer coating can be seen in STEM images (Fig. 2i) and EDS maps (Fig. 2k). Are additional carbon sources added during the recovery process? A brief discussion along these lines is required.

Response: Thanks for your suggestions. In addition to STEM images and EDS maps results, high-resolution TEM images of R-LFMP showed a uniform carbon layer coating on the particle surface with a thickness of 4–5 nm, as depicted in **Fig. R13**. **It is worth noting that no extra carbon sources were added during the regeneration process.** We attributed it to the following two reasons regarding the carbon layer coating. **First, the S-LFP and S-LMO black mass contained residual conductive carbon**

(acetylene black or carbon nanotubes) and binder (PVDF), which did not react with the DES and then precipitated along with the oxalate precursor. Subsequently, the conductive carbon and binder were refined and homogenized during the ball milling process. These compounds served as carbon sources in the LFMP synthesis process. Second, the oxalate precursor itself can also serve as a carbon source, which decomposed into the transition metal oxides and carbon during the heat treatment processes. Therefore, we believe that the above two factors can explain the origin of the carbon coating on the surface of R-LFMP. The utilization of all components in waste batteries is key to realizing sustainable recycling, which has a positive impact on resources conservation and environmental protection. The upcycling strategy of mixed cathodes in this paper is of great significance for approaching this goal.

Fig. R13 | **a** TEM image, **b-e** HRTEM images, **f** SAED pattern, **g** Enlarged regions in **I**, **h** Line profiles in **V** of R-F5M5.

3. The choice of Fe and Mn mole ratio seems to be an important factor, which is related to the performance. Please provide a detailed explanation about how to

exactly control the ratios in the R-LFMP materials.

Response: Thanks for your comments. The control of the Fe/Mn molar ratio is crucial for the regenerated materials in the upcycling processes. As for this important concern, we first re-explained it in the Methods part and the main body part, respectively, in order to show a clear description of this study for the readers.

On page 9 in the revised MS:

“Due to the above upcycling strategy, the DES system (ChCl-OA) had high leaching efficiencies (> 97%) for all elements in black mass, and the subsequent precipitation process of Fe/Mn precursors was also demonstrated, enabling that the molar ratios of Fe/Mn can be precisely controlled in the regenerated materials...”

On pages 19 and 20 in the revised MS (Methods parts):

“S-LFP and S-LMO black mass with a specific Fe/Mn molar ratio were added to DES and the mixture was then heated at 110 °C under continuous stirring for 6 h...”

“Notably, the Fe/Mn molar ratios in the precursors can be precisely controlled in black mass due to the high leaching efficiencies for all elements, which has been demonstrated in Results part...”

“The experimental conditions were the same as for the LFMP materials with different molar ratios of Fe/Mn...”

We further illustrate the control of the Fe/Mn molar ratio by combining the relevant experimental data, that is, if the Fe/Mn molar ratio is accurate in the black mass, the ratio is ideal in the precursor and LFMP materials. This is because the DES system used for the upcycling strategy has high leaching efficiencies for both S-LFP and S-LMO cathodes, as shown in **Fig. R1**.

Fig. R1 | The leaching rates of Li/P/Fe/Mn based on ICP results.

In addition, the subsequent precipitation process of Fe/Mn has displayed in **Fig. R4**.

Our strategy also shows a high separation efficiency of all elements. 89.6% of the Fe and 82.2% of the Mn were precipitated in the solid solution precursor. While 96.7% of the Li and 97.0% of the P remained in the filtrate, which were then recycled as a lithium salt (Li_3PO_4).

Fig. R4 | **d** Demonstration of the formation of DES. **e** Demonstration of the formation of the precursor. **f** FT-IR spectra of original and recycled DES. **g** The separation efficiencies of Li/Fe/Mn/P based on ICP results.

Based on above discussion, the Fe/Mn molar ratio in the upcycling strategy is controllable. Now we believe that the revised MS will provide a clear description on the main results of this study.

4. For the TEA analysis in Fig. 4, the upcycle process has a higher cost compared

with Pyro, Hydro, Direct processes. What could be the reason?

Response: Thanks for your question. In the cost analysis (Fig. R14), annualized capital cost is a critical factor in all recycling processes. Material cost is relatively lower in the Pyro process because the energy required for smelting is produced by the battery components, such as graphite. In comparison, material cost is a significant contributor to the overall cost in the other processes, such as acid/alkaline reagents in Hydro process and lithium salts in Direct and Upcycle processes. The material cost accounts for 71.1% of the total in the Upcycle process, mainly attributing to the lithium salt used for R-LFMP synthesis. Revenue analysis is directly related to the value of products. In contrast, the regenerated materials (LFMP) are the major contributors to the overall revenue in the Upcycle process. In addition, the recycled lithium salt and Mn salts also contribute to the revenue in the Upcycle process. Therefore, our proposed Upcycle process has the highest profit (profit = revenue - cost), that is 4.43 \$ per kg of feedstock.

Fig. R14 | Techno-economic analysis results of the Pyro, Hydro, Direct, and Upcycle processes. b Cost analysis. **c** Cost percentages of the Upcycle process.

Reviewer #3 (Revision marked with red in the revised MS and SI)

This paper explains a new recycling route for the cathode of a lithium ion battery. The novelty is in dissolving 2 types of cathode material (lithium iron phosphate), termed LFP, and lithium manganese oxide (LiMn_2O_4), termed LMO, to produce lithium iron manganese phosphate, termed LFMP. The use of LFMP as a promising cathode material has been known to the literature for about 1 decade as noted by the authors. The novelty of the paper is the fact that the authors have come up with a processing route that uses a so-called deep eutectic solvent (DES) to manufacture LFMP.

Response: Thanks for your helpful comments on our manuscript. With respect to your important concerns and suggestions, we have added experiments and characterizations to address them. After careful check and consideration, we have done substantial revisions in the revised manuscript (MS) and supplementary information (SI). **All the changes were highlighted with red in the revised MS and SI.** Below are the point-to-point response to your comments.

The mixture of many chemicals used to react with the 2 cathode materials are given in the methods section without any justification. For example, carbon nanotubes are used without explanation.

Response: Thanks for your comments. We carefully checked all the materials and chemicals involved to ensure that they were useful in the subsequent experimental

processes. Specifically, we redefined the abbreviations for some chemicals, such as **choline chloride** ($C_5H_{14}ClNO$, ChCl, 98%), **oxalic acid** ($H_2C_2O_4$, OA, 99%). ChCl and OA were used for preparing the DES in this work. Ethanol was used as the ball milling medium during the synthesis of R-LFMP.

On pages 19 in the revised MS (Methods parts):

*“...Spent LCO cathode powder was separated from a mobile phone battery. Chemicals involve **choline chloride** ($C_5H_{14}ClNO$, ChCl, 98%), **oxalic acid** ($H_2C_2O_4$, OA, 99%), **lithium hydroxide** (LiOH, 99%), **ammonium dihydrogen phosphate** ($NH_4H_2PO_4$, 98%), **FeC₂O₄** (99%), **MnC₂O₄·2H₂O** (99%), **ethanol** (98%) ...”*

Furthermore, we gave a detailed explanation about the chemicals and materials used in the Methods section. Acetylene black (AB) and carbon nanotubes (CNT) were used as the conductive carbon, which is the consensus for manufacturing the cathodes. Specially, phosphate materials have the lowest electrical conductivity among all cathode materials⁵⁻⁸, making it important to add highly conductive materials for improving the electronic conductivity, such as AB and CNT.

In addition, electronic conductivity measurements were carried out by Four-point probe. Commercial LFP (C-LFP) and LFMP (C-LFMP) materials were compared with our regenerated LFMP, and each sample was tested six times (**Table R1** and **Fig. R15**). C-LFP has the best electrical conductivity. With the increase of Mn content, the conductivity showed a decreasing trend, which was consistent with the literature reports. We believe that above discussion could explain the reasons of carbon nanotubes used in electrode preparation processes.

Table R1 | Electronic conductivity results for C-LFP, R-F8M2, R-F5M5, and C-LFMP

samples. (Each sample was tested six times. Unit: S cm^{-1} .)

	1	2	3	4	5	6
C-LFP	10.2322	10.1631	10.2181	10.1994	10.2009	10.1838
R-F8M2	1.5883	1.5878	1.5878	1.5882	1.5878	1.5887
R-F5M5	0.0642	0.0716	0.0712	0.0704	0.0682	0.0713
C-LFMP	0.0021	0.0022	0.0021	0.0022	0.0021	0.0022

Fig. R15 | Electronic conductivity results for C-LFP, R-F8M2, R-F5M5, and C-LFMP samples.

Finally, we have supplemented the experiments to reveal the critical effect of CNT in LFMP cathodes. In electrochemical performance tests (**Fig. R16**), R-LFMP cathodes with 10wt % and 5wt% CNT had the similar capacity and cycling capability. As comparison, R-LFMP cathodes with 0wt% CNT had very poor properties, suggesting that CNT plays a critical role in performance of R-LFMP, but does not provide additional capacity contributions.

Fig. R16 | The influence of CNT on electrochemical performance of R-F5M5. a-c Initial two charge and discharge curves at 0.1C. **d-f** Cycling performance at 1C rate.

The authors show theoretically that the processing route is environmentally friendly and is cost effective. The authors have manufactured and characterised two recycled forms of LFMP which they describe as R-F5M5 and R-F8M2. They do not explain the differences in manufacturing route for each.

Response: Thanks for your comments. The manufacturing route for R-F5M5 and R-F8M2 was totally the same since the upcycling approach in this work is universal for synthesizing LFMP with different Fe/Mn ratios. For a clear description of this study for the readers, we re-explained it in the Methods section and the main body, respectively.

On page 9 in the revised MS:

“Due to the above upcycling strategy, the DES system (ChCl-OA) had high leaching efficiencies (> 97%) for all elements in black mass, and the subsequent precipitation process of Fe/Mn precursors was also demonstrated, enabling that the molar ratios of Fe/Mn can be precisely controlled in the regenerated materials...”

On pages 19 and 20 in the revised MS (Methods section):

“S-LFP and S-LMO black mass with a specific Fe/Mn molar ratio were added to DES and the mixture was then heated at 110 °C under continuous stirring for 6 h...”

“Notably, the Fe/Mn molar ratios in the precursors can be precisely controlled in black mass due to the high leaching efficiencies for all elements, which has been demonstrated in Results part...”

“The experimental conditions were the same as for the LFMP materials with different molar ratios of Fe/Mn...”

Given that the Fe/Mn molar ratio is accurate in the black mass, the ratio is ideal in the precursor and LFMP materials. This is because the DES system used for the upcycling strategy has high leaching efficiencies for both S-LFP and S-LMO cathodes.

Lots of detail on the chemical structure of R-F5M5 is given in Fig. 2, but there is almost no information given on R-F8M2. The electrochemical performance of the two alloys are compared in Fig. 3. The paper is imbalanced in this regard.

Response: Thanks for your suggestions. We have supplemented the related characterizations for R-F8M2 including XRD refinement, in-depth XPS, TEM. As shown in **Fig. R17** and **Table R2**, R-F8M2 has an ordered *Pnma* space group and olivine crystal structure. The cell volume (296.78 \AA^3) is smaller than that of R-F5M5 (296.83 \AA^3) due to Mn having a larger atomic radius than Fe.

Fig. R17 | XRD pattern and Rietveld refinement results for R-F8M2.

Table R2 | The XRD refinement results for R-F8M2.

LiMn_{0.2}Fe_{0.8}PO₄ (Space Group Pnma)						
	Atom	x	y	z	Occ.	Biso.
Atomic Occupancies	Li	0.0000	0.0000	0.0000	1.0	0.6226(2)
	Fe	0.2820(1)	0.2500	0.9731(9)	0.8	2.0686(8)
	Mn	0.2820(1)	0.2500	0.9731(9)	0.2	
	P	0.0963(1)	0.2500	0.4261(1)	1.0	2.1677(4)
	O ₁	0.0874(1)	0.2500	0.7460(2)	1.0	2.2195(2)
	O ₂	0.4543(1)	0.2500	0.2222(2)	1.0	1.8578(8)
	O ₃	0.1591(5)	0.0519(4)	0.2798(8)	1.0	1.9197(5)
Lattice Parameters	a / Å	b / Å		c / Å	V / Å³	
	10.3415(3)	6.0208(5)		4.7021(9)	296.78(2)	
Agreement Factors						
χ^2	1.41%		R_p	1.27%		R_{wp} 1.02%

Depth XPS profiling for R-F8M2 showd uniform distributions of Fe and Mn from the surface to the bulk (**Fig. R18**). Atomic concentration of Fe was higher than that of Mn with the increasing of etching time. Specifically, the peaks at 710.1 eV and 641.7 eV are respectively ascribed to Fe $2p_{3/2}$ and Mn $2p_{3/2}$, suggesting that the valence states of Fe and Mn are +2 in R-F8M2.

Fig. R18 | Depth etching XPS results of R-F8M2. a Atomic concentrations, **b** Fe 2p XPS, **c** Mn 2p XPS.

Furthermore, the microstructure of R-F8M2 was revealed by TEM and HRTEM images, as displayed in **Fig. R19**. The average particle size was about 100 nm and a uniform carbon layer coated on the particle surface with a thickness of 3–4 nm. HAADF-STEM images showed a clear lattice fringe (0.31nm), which was ascribed to the (121) interplanar spacing. Qualitative analysis of EDS results indicated that the elemental contents were consistent with R-F8M2, and Fe, Mn, P, O, and C elements distributed uniformly through the particles. **The characterizations on the chemical structure of R-F8M2 have been added in revised MS and SI (on pages 9, 10, 13, 26). Now we believe the current version is reasonable in the overall structure, which will provide a clear description for the readers.**

Fig. R19 | Microstructure characterizations of R-F8M2. **a** TEM image, **b-c** HRTEM images, **d** SAED pattern in **b**, **e** Line profiles in **c**, **f** EDS elemental contents, **g** EDS maps.

The battery properties of LFMP are an improvement upon those of LFP and LMO, but are not outstanding.

Response: Thanks for your comments. LFMP is a structural upgrade product based on the LFP, that is, substituting a transition metal (Mn) at Fe²⁺ sites in LFP to form a LiFe_xMn_{1-x}PO₄ solid solution. In comparison, an improvement on working potential is beneficial to increasing the energy density given that the specific capacity is almost the same. R-F5M5 has a mean voltage (3.68 V versus Li/Li⁺) and a specific energy of 559 Wh kg⁻¹ which is higher than that of a commercial LFP (C-LFP) cathode material (3.38V and 524 Wh kg⁻¹) (**Fig. R20**). This work focuses on providing a sustainable way for recycling mixed cathode materials. Also, the value-added of recovered products is

also an important consideration.

Fig. R20 | b Comparison of the discharge capacities, middle voltages, and energy densities. **c** Middle voltage retention at 1C rate after 200 cycles.

The paper is highly detailed in its electrochemical results, and it is not clear to the reviewer that this will appeal to the general audience.

Response: Thanks for your comments. The electrochemical results are detailed since the difference in performance is an important basis for us to evaluate the battery material, such as voltage platform, capacity, cycle stability and so on. In this paper, degraded LFP cathodes were upgrade into LFMP cathodes with an increased voltage and energy density. In addition, the upcycling concept demonstrated in this paper is one of the hot topics in the field of battery recycling, which realizes the value-added utilization of recycled products.

For a clear description of the recycling mechanism, we have added a schematic and the corresponding discussion in **Fig. 1b** and revised MS, respectively. This supplement can better explain the structural recombination of cathode materials and transition metal rearrangement in the regeneration process, thus increasing theoretical depth for this

paper. We have added a more comprehensive discussion on the scale-up strategy for practical application of the proposed upcycling process, as shown in **Figs. R9-10**, demonstrating that it has the potential for practical large-scale applications. In addition, we expand the idea on the basis of the upcycling strategy, that is, a potential recycling approach for LFMP was proposed and verified by experiments (*see next response*). Hence, we believe that the interesting strategy and extended discussion in this paper will appeal to the general audience.

Fig. R9 | Pictures for small scale-up experiments in the lab.

Fig. R10 | **a** XRD patterns for the precursor and R-F5M5 synthesized by the scale-up experiments. **b** Electrochemical performance of R-F5M5-scale sample.

There is no mention on how the authors propose to recycle the LFMP.

Response: Thanks for your comments. We expand the idea on the basis of the upcycling strategy, that is, **a potential recycling approach for LFMP was proposed and verified by experiments**. LFMP has the same olivine type crystal structure as LFP and can be dissolved by the DES (ChCl-OA). As shown in **Fig. R11**, here we used the commercial LFMP (C-LFMP) and regenerated LFMP (R-LFMP) as the raw materials, respectively. The same target product was formed, that is, the iron/manganese oxalate precursor, suggesting that the upcycling approach in this work is also applicative to process LFMP cathodes. Based on above discussion, **a sustainable recycling system, involving the upcycling strategy in this work and potential LFMP recycling in the future**, as summarized in **Fig. R12**. LFMP is a promising cathode material with an increased voltage and energy density. Especially in recent years, many battery manufacturers began to layout LFMP batteries. We believe that LFMP will usher in explosive growth in the next few years, so it can be inferred that sustainable recycling of LFMP batteries will also be a hot topic.

Fig. R11 | **a** Picture for C-LFMP and R-LFMP dissolved in the DES. **b** XRD patterns for the obtained precursors.

Fig. R12 | Schematic for the upcycling strategy in this work and the potential LFMP recycling in the future.

Finally, we have added an in-depth discussion on the practical applications of this approach, as shown in revised MS.

On page 19 in the revised MS (Discussion section):

“Considering that the potential feasibility for practical application is crucial for evaluating a battery recycling process, we carried out a scalable experiment demonstration (Supplementary Fig. 26). As expected, the experimental phenomenon was consistent with the previous processes, and the R-LFMP samples synthesized from the scale-up experiments showed satisfying properties (Supplementary Fig. 27), demonstrating that the upcycling approach was able to expanding at a gram-grade scale. Furthermore, we have expanded the scope of practical applications of the upcycling strategy, that is, a potential recycling approach for LFMP was proposed and verified by experiments (Supplementary Fig. 28). A sustainable recycling system, involving the upcycling strategy in this work and potential LFMP recycling in the future, is summarized in Fig. 5. LFMP, regarded as the structural upgrade product of LFP, has attracted much attention in both academic and industrial fields in recent years. The DES system in this work was also applicable to process LFMP, suggesting that the proposed upcycling strategy can achieve a closed-loop recycling between the mixed spent cathode chemistries and the next-generation batteries.”

The English is poor in some places. For example, the authors frequently use the word ‘even’ to mean ‘uniform’.

Response: Thanks for your suggestions. We have corrected the improper words such as ‘even’. Also, we have polished the language of the paper with the help of an English native speaker.

References

1. Chang, X., *et al.* Selective extraction of transition metals from spent $\text{LiNi}_x\text{Co}_y\text{Mn}_{1-x-y}\text{O}_2$ cathode via regulation of coordination environment. *Angew. Chem. Int. Ed.* **61**, e202202558 (2022).
2. Wang, M. M., *et al.* Selective extraction of critical metals from spent lithium-ion batteries. *Environ. Sci. Technol.* **57**, 3940-3950 (2023).
3. Lu, Q. Q., *et al.* Sustainable and convenient recovery of valuable metals from spent Li-ion batteries by a one-pot extraction process. *ACS Sustain. Chem. Eng.* **9**, 13851-13861 (2021).
4. Thompson, D. L., Pateli, I. M., Lei, C. H., Jarvis, A., Abbott, A. P. & Hartley, J. M. Separation of nickel from cobalt and manganese in lithium ion batteries using deep eutectic solvents. *Green Chem.* **24**, 4877-4886 (2022).
5. Manthiram, A. A reflection on lithium-ion battery cathode chemistry. *Nat. Commun.* **11**, 1550 (2020).
6. Molenda, J., Ojczyk, W. & Marzec, J. Electrical conductivity and reaction with lithium of $\text{Li}_{1-y}\text{MnPO}_4$ olivine-type cathode materials. *J. Power Sources* **174**, 689-694 (2007).
7. Park, S. Y., *et al.* Probing electrical degradation of cathode materials for lithium-ion batteries with nanoscale resolution. *Nano Energy* **49**, 1-6 (2018).
8. Park, N. Y., Park, G. T., Kim, S. B., Jung, W., Park, B. C. & Sun, Y. K. Degradation mechanism of Ni-rich cathode materials: Focusing on particle interior. *ACS Energy Lett.*

7, 2362-2369 (2022).

REVIEWERS' COMMENTS

Reviewer #1 (Remarks to the Author):

The raised concerns have been effectively addressed. The current version of the manuscript is ready for publication without the necessity for additional revisions.

Editorial note: Reviewer #1 was additionally asked to comment in the place of Reviewer #3. The additional comments are as follows:

Regarding the revision based on the comment from Reviewer #3, The manuscript has been revised accordingly in response to the comments. I agree to its publication.

Reviewer #2 (Remarks to the Author):

The author has made changes according to the reviewer's suggestions, and this version can be accepted.

Point-by-point response to reviewers' comments

Reviewer #1:

The raised concerns have been effectively addressed. The current version of the manuscript is ready for publication without the necessity for additional revisions.

Response: Thanks again for your constructive comments on this manuscript.

Reviewer #1 was additionally asked to comment in the place of Reviewer #3. The additional comments are as follows:

Regarding the revision based on the comment from Reviewer #3, The manuscript has been revised accordingly in response to the comments. I agree to its publication.

Response: We thank the referee for his/her positive evaluation.

Reviewer #2:

The author has made changes according to the reviewer's suggestions, and this version can be accepted.

Response: We appreciate your comments to improve the quality of our manuscript.